# MT3: Multi-Task Multitrack Music Transcription

**Josh Gardner,**[*] **Ian Simon, Ethan Manilow,**[†] **Curtis Hawthorne, Jesse Engel**
Google Research, Brain Team

## Abstract

Automatic Music Transcription (AMT), inferring musical notes from raw audio, is a challenging task at the core of music understanding. Unlike Automatic Speech Recognition (ASR), which typically focuses on the words of a single speaker, AMT often requires transcribing multiple instruments simultaneously, all while preserving fine-scale pitch and timing information. Further, many AMT datasets are "low-resource", as even expert musicians find music transcription difficult and time-consuming. Thus, prior work has focused on task-specific architectures, tailored to the individual instruments of each task. In this work, motivated by the promising results of sequence-to-sequence transfer learning for low-resource Natural Language Processing (NLP), we demonstrate that a general-purpose Transformer model can perform multi-task AMT, jointly transcribing arbitrary combinations of musical instruments across several transcription datasets. We show this unified training framework achieves high-quality transcription results across a range of datasets, dramatically improving performance for low-resource instruments (such as guitar), while preserving strong performance for abundant instruments (such as piano). Finally, by expanding the scope of AMT, we expose the need for more consistent evaluation metrics and better dataset alignment, and provide a strong baseline for this new direction of multi-task AMT.[1]

## 1 Introduction

Recorded music often contains multiple instruments playing together; these multiple "tracks" of a song make music transcription challenging for both algorithms and human experts. A transcriber must pick each note out of the audio mixture, estimate its pitch and timing, and identify the instrument on which the note was performed. An AMT system should be capable of transcribing multiple instruments at once (Multitrack) for a diverse range of styles and combinations of musical instruments (Multi-Task). Despite the importance of Multi-Task Multitrack Music Transcription (MT3), several barriers have prevented researchers from addressing it. First, no model has yet proven capable of transcribing arbitrary combinations of instruments across a variety of datasets. Second, even if such models existed, no unified collection of AMT datasets has been gathered that spans a variety of AMT tasks. Finally, even within current AMT datasets, evaluation is inconsistent, with different research efforts using different metrics and test splits for each dataset.

Compounding this challenge, many music datasets are relatively small in comparison to the datasets used to train large-scale sequence models in other domains such as NLP or ASR. Existing open-source music transcription datasets contain between one and a few hundred hours of audio (see Table 1), while standard ASR datasets LibriSpeech (Panayotov et al., 2015) and CommonVoice (Ardila et al., 2020) contain 1k and 9k+ hours of audio, respectively. LibriSpeech alone contains more hours of audio than all of the AMT datasets we use in this paper, combined. Taken as a whole, AMT fits the general description of a "low-resource" task, where data is scarce.

In this work, we provide a strong empirical contribution to the field by overcoming each of these barriers and enabling Multi-Task Multitrack Music Transcription (MT3). Our contributions include:

---

[*]Paul G. Allen School of Computer Science & Engineering. Work performed as a Google Research intern.

[†]Interactive Audio Lab, Northwestern University. Work performed as a Google Student Researcher.

[1]We encourage reviewers to view more extensive transcription results, including audio examples, at `https://storage.googleapis.com/mt3/index.html`.

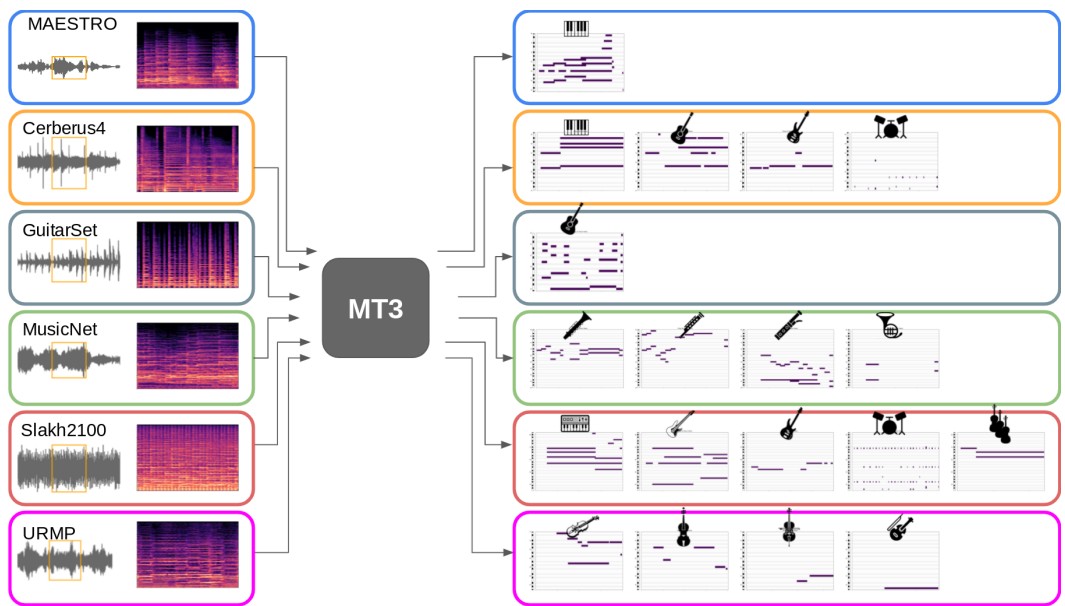

Figure 1: MT3 is capable of transcribing an arbitrary number of instruments from raw audio spectrograms. Shown here are real 4-second audio clips, pianorolls reconstructed from the model's tokenized output, and the corresponding instrument labels (additional Slakh2100 instruments omitted due to space). Note that in some cases, multiple notes predicted from a monophonic instrument (such as clarinet or French horn) reflects an ensemble containing multiple players of that instrument.

**Unified framework for training:** We define a tokenization scheme with a compact and flexible vocabulary to convert between model output tokens and multitrack MIDI files, enabling a sequence-to-sequence approach inspired by Raffel et al. (2019) and Xue et al. (2020) that supports datasets with different combinations of instruments. This allows us to simultaneously leverage several datasets which were previously only used in isolation due to differences in instrumentation.

**Benchmark collection of diverse datasets:** We assemble six multitrack AMT datasets, spanning a variety of dataset sizes, styles, and instrumentations. Together they form the largest known collection publicly available for multi-task AMT training.

**Consistent evaluation:** We define standard test set splits and apply a consistent set of note-based metrics across all six datasets. We also introduce a new instrument-sensitive transcription metric to jointly evaluate note and instrument accuracy.

**SOTA Baseline:** Training an off-the-shelf T5 architecture with our framework, we realize strong baseline models that achieve SOTA transcription performance on each individual multitrack dataset, outperforming prior dataset-specific transcription models as well as professional-quality DSP-based transcription software. Our model, which we refer to as MT3, demonstrates very high instrument labeling accuracy across all six datasets, even when many instruments are simultaneously present, and is robust to the grouping of instruments.

**Improving low-resource AMT:** By training a *single model* across a mixture of all six datasets, we find that MT3 performance dramatically improves for low-resource datasets over the baseline models (up to 260% relative gain), while preserving strong performance on high-resource datasets.

## 2 RELATED WORK

### 2.1 TRANSFORMERS FOR SEQUENCE MODELING

The Transformer architecture, originally proposed in Vaswani et al. (2017), has recently demonstrated strong performance across many sequence modeling tasks in several domains. For example, T5 (Raffel et al., 2019) demonstrated that many language tasks previously addressed with separate

models could be addressed using a single text-to-text encoder-decoder Transformer model. Extending this approach, mT5 (Xue et al., 2020) used a single Transformer to model multiple languages, demonstrating that a unified architecture could also serve as a general multilingual model, leveraging high-resource language datasets to improve model performance on lower-resource datasets. Other prominent examples of Transformer-based architectures for sequence modeling include BERT (Devlin et al., 2018) and the GPT family of models, most prominently GPT-3 (Brown et al., 2020).

Transformers have also been applied to some audio modeling tasks. For example, Transformer-based models have been used for audio classification (Gong et al., 2021; Verma & Berger, 2021), captioning (Mei et al., 2021), compression (Dieleman et al., 2021), speech recognition (Gulati et al., 2020), speaker separation (Subakan et al., 2021), and enhancement (Koizumi et al., 2021). Transformers have also been used for generative audio models (Dhariwal et al., 2020; Verma & Chafe, 2021), which in turn have enabled further tasks in music understanding (Castellon et al., 2021).

## 2.2 MUSIC TRANSCRIPTION

Historically, music transcription research has focused on transcribing recordings of solo piano (Poliner & Ellis, 2006; Böck & Schedl, 2012; Kelz et al., 2016). As a result, there are a large number of transcription models whose success relies on hand-designed representations for piano transcription. For instance, the Onsets & Frames model (Hawthorne et al., 2017) uses dedicated outputs for detecting piano onsets and the note being played; Kelz et al. (2019) represents the entire amplitude envelope of a piano note; and Kong et al. (2020) additionally models piano foot pedal events (a piano-specific way of controlling a note's sustain). Single-instrument transcription models have also been developed for other instruments such as guitar (Xi et al., 2018) and drums (Cartwright & Bello, 2018; Callender et al., 2020), though these instruments have received less attention than piano.

Although not as widespread, some multi-instrument transcription systems have been developed. For example, Manilow et al. (2020) presents the Cerberus model, which simultaneously performs source separation and transcription for a fixed and predefined set of instruments. Lin et al. (2021) also perform both separation and transcription, albeit based on an audio query input and with the addition of synthesis. ReconVAT (Cheuk et al., 2021) uses an approach based on U-Net and unsupervised learning techniques to perform transcription on low-resource datasets; however, the model does not predict instrument labels, instead outputting a single pianoroll that combines all instruments into a single "track". A similar limitation applies to the early transcription system introduced alongside the MusicNet dataset by Thickstun et al. (2016). Tanaka et al. (2020) uses a clustering approach to separate transcribed instruments, but the model output does not include explicit instrument labels. In contrast, our model outputs a stream of events representing notes from an arbitrary number of instruments with each note explicitly assigned to an instrument; it learns to detect the presence (or absence) of instruments directly from audio spectrograms (see Figure 1).

The work most closely related to ours is Hawthorne et al. (2021), which uses an encoder-decoder Transformer architecture to transcribe solo piano recordings. Here, we extend their approach to transcribe polyphonic music with an arbitrary number of instruments. We adhere to their philosophy of using components close to "off-the-shelf" as possible: spectrogram inputs, a standard Transformer configuration from T5, and MIDI-like output events.

## 3 TRANSCRIPTION MODEL

At its core, music transcription can be posed as a sequence-to-sequence task, where the input is a sequence of audio frames, and the output is a sequence of symbolic tokens representing the notes being played. A key contribution of this work is to frame *multi-instrument* transcription, where different source instruments are present in a single input audio stream, within this paradigm and allow the model to *learn* which instruments are present in the source audio using Transformer model paired with a novel vocabulary designed to support this general task.

### 3.1 TRANSFORMER ARCHITECTURE

A key contribution of our work is the use of a single generic architecture — Transformers via T5 (Raffel et al., 2019) — to address a variety of tasks previously tackled using complex, handcrafted,

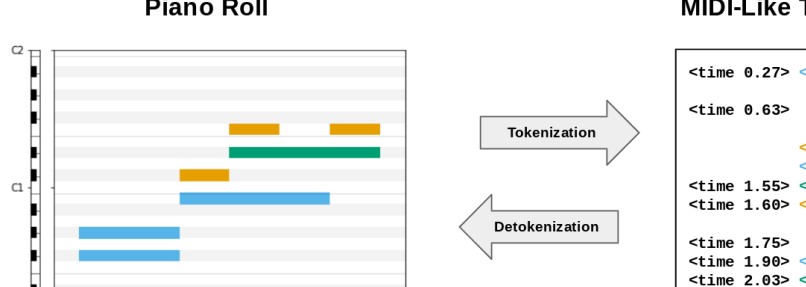

Figure 2: Tokenization/detokenization, as described in Section 3.2. MIDI data (left, represented here as a multitrack "pianoroll") can be tokenized into MIDI-like target tokens for training (right). Output tokens using the same vocabulary can be deterministically decoded back into MIDI data.

dataset-specific architectures. The T5 architecture is an encoder-decoder Transformer model which closely follows the original form in Vaswani et al. (2017).

In the T5 architecture, a sequence of inputs is mapped to a sequence of learned embeddings plus fixed positional embeddings; we use absolute positional embeddings instead of the bucketed "relative" embeddings used in Raffel et al. (2019) to ensure that all positions can be attended to with equal resolution. The model uses a series of standard Transformer self-attention "blocks" in both the encoder and decoder. In order to produce a sequence of output tokens, the model uses greedy autoregressive decoding: an input sequence is fed in, the output token with the highest predicted probability of occurring next is appended to the sequence, and the process is repeated until an end-of-sequence (EOS) token is produced. We use the T5 "small" model, which contains approximately 60 million parameters. While much larger models are commonly used in language modeling tasks, we found that increasing model size tended to exacerbate overfitting. Full details on the Transformer architecture used in this work are given in Appendix A. Additionally, we make our code available along with the release of this paper at `https://github.com/magenta/mt3`.

## 3.2 MODEL INPUTS AND OUTPUTS

As shown in Figure 1, MT3 uses log Mel spectrograms as inputs. For the outputs, we construct a token vocabulary inspired by the MIDI specification, which we refer to as "MIDI-like" because it contains a subset of the original MIDI[2] specification (1996) (e.g. our vocabulary does not represent "control change" MIDI events). This output is a modification of the vocabulary of Hawthorne et al. (2021) with the following differences: (1) addition of instrument change tokens, which allow for multiple instruments to be represented in a single event stream; (2) removal of velocity, as most of our training datasets do not contain velocity annotations (and there is no standardized method for coding velocity across datasets); (3) "ties" to better handle notes that span multiple segments. The vocabulary, illustrated in Figure 2, consists of the following token types:

**Instrument (128 values):** Indicates which instrument the following messages should be directed to. The specified instrument will be used for all subsequent events until the next Instrument event. The choice of 128 distinct values is selected to match the original General MIDI specification, which contains 128 "programs" used to designate specific instruments. We further discuss the challenges of representing instruments using program numbers below.

**Note (128 values):** Represents a note-on or note-off event for one of the 128 MIDI pitches.

**On/Off (2 values):** Changes whether subsequent Note events are interpreted as note-on or note-off.

**Time (205 values):** Indicates the absolute time location of one or more events within a segment, quantized to 10 ms intervals. This time applies to all subsequent events until the next Time event, which allows for an arbitrary number of notes to occur at a given time point. Time events must occur

---

[2]MIDI (Musical Instrument Digital Interface) is an industry standard music technology protocol used to represent musical data and allow communication between musical devices. (`https://www.midi.org/`)

in chronological order. (The number of possible Time values depends on the Transformer's input sequence length, which is 2.048 seconds for all of our experiments.)

**Drum (128 values):** Represents a drum onset from one of 128 drum types in the General MIDI standard. Drums are not a focus of this work, but we include them for completeness.

**End Tie Section (1 value):** Ends the "tie" section at the beginning of a segment (see below).

**EOS (1 value):** Used to indicate the end of a sequence.

A key contribution of this work is the demonstration that this highly general and flexible output vocabulary can be used to learn a single transcription model that works well across different instruments, datasets, and orchestrations, without manual tuning of the model or vocabulary. In contrast, prior works have been limited to models which transcribe only a single instrument (Hawthorne et al., 2021), contain separate transcription "heads" for a fixed set of instruments (Manilow et al., 2020), or ignore the instrument dimension entirely and only transcribe the notes (Cheuk et al., 2021).

One limitation of sequence models applied to audio is that most audio sequences are too large to fit in memory when modeling using a Transformer architecture, which requires $\mathcal{O}(n^2)$ memory with respect to sequence length for the self-attention blocks. In order to address these constraints, we use the procedure described by Hawthorne et al. (2021): audio is split into smaller, non-overlapping segments, with input spectrograms and event tokens extracted from each segment. The model processes each segment *independently*. The same procedure is used for both training and inference; however, at inference time we take the additional step of concatenating the decoded events from all segments into a single sequence to reconstruct the full example.

One issue with transcribing cropped audio segments independently is that a note may span multiple segments. While our basic approach is often able to handle such cases, we find that occasionally the model will forget to "turn off" a note. To ameliorate this problem, we introduce a "tie" section at the beginning of each segment where the model must declare which notes are already active; that is, the model is trained to emit Program and Pitch tokens for already-active notes, followed by the End Tie Section token, followed by the events of the segment. When concatenating segments to reconstruct an entire transcribed example, we end any notes not explicitly declared in the tie section. This allows the model to fail gracefully when it detects a note-on in one segment but not the corresponding note-off in a subsequent segment.

### 3.3 MULTI-TASK MIXTURE

In addition to removing the cumbersome task of constructing specialized architectures and loss functions for different instrumentations and datasets, our general output vocabulary also allows our model to be trained on a mixture of several datasets simultaneously, similar to how multilingual translation models such as mT5 are trained on several languages (Xue et al., 2020). This approach not only simplifies model design and training, but also increases the amount and diversity of training data available to the model. As noted previously, scarcity of training data has been a major challenge for prior AMT modeling efforts. This mixture approach has not previously been demonstrated in the music transcription literature; instead, prior works have often focused on training separate models for individual datasets (i.e. Cheuk et al. (2021)). We note that "mixing" here refers to including data from multiple datasets within a single training batch.

In order to balance model performance on low- and high-resource datasets, we use a temperature sampling strategy for the mixing as follows: if dataset $i$ has $n_i$ examples, we sample an example from that dataset with probability $(n_i \,/\, \sum_j n_j)^{0.3}$, similar to mT5 (Xue et al., 2020). This has the effect of increasing the frequency with which the model observes examples from low-resource datasets during training, while observing examples from high-resource datasets with lower frequency.

## 4 EXPERIMENTS

We conduct a series of experiments to test our approach. In particular, we evaluate the overall transcription quality of our model across six datasets including both high- and low-resource datasets, evaluate the effect of instrument groupings, and use our results to identify labeling issues with certain

| Dataset | Hrs. Audio | Num. Songs | Num. Instr. | Instr. Per Song | Align | Low-Resource | Synthetic | Drums |
|---|---|---|---|---|---|---|---|---|
| **Slakh2100** | 969 | 1405 | 35 | 4–48 | Good | | ✓ | ✓ |
| **Cerberus4** | 543 | 1327 | 4 | 4 | Good | | ✓ | ✓ |
| **MAESTROv3** | 199 | 1276 | 1 | 1 | Good | | | |
| **MusicNet** | 34 | 330 | 11 | 1–8 | Poor | ✓ | | |
| **GuitarSet** | 3 | 360 | 1 | 1 | Good | ✓ | | |
| **URMP** | 1 | 44 | 14 | 2–5 | Fair | ✓ | | |

Table 1: Datasets used in this paper.

datasets. We also assess our models' generalization to out-of-domain data with a series of leave-one-dataset-out experiments and use our results to identify label quality issues in Section D.

## 4.1 DATASETS

Our experiments use six datasets of varying size, recording process, instrumentation, and genre. In addition to demonstrating the flexibility of our approach, this also allows us to compare to a number of different baseline models, each of which can only be fairly applied to specific datasets, and to offer a single SOTA baseline across all datasets using MT3. The six datasets are described briefly below; we provide further information about these datasets in Table 1 and Appendix B.

**MAESTROv3:** MAESTROv3 (Hawthorne et al., 2018) is collected from a virtual classical piano competition, where audio and detailed MIDI data are collected from performers playing on Disklavier pianos that electronically capture the performance of each note in real time.

**Slakh2100:** Slakh2100 consists of audio generated by rendering MIDI files using professional-grade, sample-based synthesis software. Its construction is detailed in Manilow et al. (2019). During training, we use a form of data augmentation to combine together different subsets of the individual-instrument mixes, which we describe in Appendix B.

**Cerberus4:** Cerberus4 is derived from the Slakh2100 dataset, obtained by mixing all combinations of the four instruments (guitar, bass, drums, piano) in tracks where those instruments are active. These are also the instruments used by the 4-instrument Cerberus model of Manilow et al. (2020).

**GuitarSet:** GuitarSet (Xi et al., 2018) is composed of live guitar performances of varied genre, tempo, and style, recorded using a high-precision hexaphonic pickup that individually captures the sound of each guitar string. MIDI labels for each track are derived from these recordings.

**MusicNet:** MusicNet (Thickstun et al., 2016) consists of freely-licensed classical music recordings from a variety of instruments and ensemble types paired with human-generated transcriptions crowdsourced from expert human transcribers. Since labels are primarily aligned with dynamic time warping, they are less accurate than other datasets.

**URMP:** The University of Rochester Multi-Modal Music Performance (URMP) Dataset (Li et al., 2018) is composed of multi-instrument classical pieces with diverse instrumentation. The individual instruments are recorded separately and mixed, and the aligned MIDI labels come from human annotators who corrected $f_0$ curves derived from the pYIN (Mauch & Dixon, 2014) algorithm.

We discuss the label alignment quality of MusicNet and URMP in Appendix D.2. In the phrasing of the NLP literature, we will refer to GuitarSet, MusicNet, and URMP as "low-resource" datasets, as they contain only 3, 34, and 1.3 hours of total audio, respectively. This makes these datasets challenging to learn from — particularly MusicNet and URMP, which contain many distinct instruments with several instruments per track, as shown in Table 1.

## 4.2 EVALUATION

The metrics used to evaluate multi-instrument transcription models in the literature are inconsistent, even when ignoring the multi-instrument vs. single-instrument distinction. For example, several variants of "F1 score" can be computed using the standard library for music transcription evaluation `mir_eval` (Raffel et al., 2014), which differ based on whether note *offsets* (the time at which a

note ends) should be considered in addition to pitch values and note onsets (the time at which a note begins) in evaluating whether a prediction is correct.

To provide the fairest and most complete comparison to existing work, we evaluate models on each dataset using three standard measures of transcription performance: Frame F1, Onset F1, and Onset-Offset F1. We use the standard implementation of these metrics from the `mir_eval` Python toolkit. For each metric, `mir_eval` uses bipartite graph matching to find the optimal pairing of reference and estimated notes, then computes precision, recall, and F1 score using the following criteria:

**Frame F1** score uses a binary measure of whether a pianoroll-like representation of the predictions and targets match. Each second is divided into a fixed number of "frames" (we use 62.5 frames per second), and a sequence of notes is represented as a binary matrix of size [frames × 128] indicating the presence or absence of an active note at a given pitch and time.

**Onset F1** score considers a prediction to be correct if it has the same pitch and is within ±50 ms of a reference onset. This metric ignores note offsets.

**Onset-Offset F1** score is the strictest metric commonly used in the transcription literature. In addition to matching onsets and pitch as above, notes must also have matching offsets. The criterion for matching offsets is that offsets must be within $0.2 \cdot$ reference_duration or 50 ms from each other, whichever is greater: $|\text{offset\_diff}| \leq \max(0.2 \cdot \text{reference\_duration}, 50 \text{ ms})$ (Raffel et al., 2014).

While these three metrics are standard for music transcription models, they offer only a limited view of the performance of a *multi-instrument* model, as none consider *which instrument* is predicted to play which notes in a sequence. This is due to the facts that (1) most prior music transcription models were limited to single-instrument transcription, and (2) even most multi-instrument transcription models did not assign specific instruments to predicted notes; we also believe the lack of a true multi-instrument metric has led to this gap in prior work. Thus, we also propose and evaluate our models' performance using a novel metric which we call *multi-instrument F1*.

**Multi-instrument F1** adds to the Onset-Offset F1 score the additional requirement that the *instrument* predicted to play a note must match the instrument of the reference note. This is a stricter metric than the MV2H metric proposed in McLeod & Steedman (2018), as MV2H ignores offsets and also eliminates notes from ground-truth during evaluation of the instrument labels when the pitch is not correctly detected; multi-instrument F1 is also more directly related to the existing transcription metrics widely used in prior works.

Due to the limitations of previous models, it is often not possible to compute a multi-instrument F1 score for previous works; as a result, we only provide this metric for our model (shown in Table 3).

We also evaluate our models' ability to transfer to unseen datasets by conducting "leave-one-dataset-out" (LODO) training experiments. These results demonstrate the generality of our approach in transferring to entirely new datasets, and are also useful in evaluating the impact of the various datasets used on the final model performance.

Two of our datasets (Slakh2100, Cerberus4) contain drums. When evaluating our model, we match reference and estimated drum hits using onset time and General MIDI drum type (as the concept of a "drum offset" is not meaningful); a more rigorous evaluation methodology focused specifically on drums can be found in Callender et al. (2020).

### 4.2.1 BASELINES

For each dataset, we compare to one or more baseline models. In addition to comparing to previous works which developed machine learning models for multi-instrument transcription on one or more of our datasets, we also compare our results to a professional-quality DSP software for polyphonic pitch transcription, Melodyne[3]. Details on our usage of Melodyne are provided in Section C.

As a consequence of the dataset-specific training and architectures mentioned above, not all models are appropriate for all datasets. As a result, we only provide results for baseline models on datasets containing the instruments for which the original model was designed: for example, we evaluate the Cerberus model (Manilow et al., 2020) only on the Cerberus4 dataset, which contains the four instruments for which the model contains specific transcription heads, and GuitarSet, where we

---

[3]https://www.celemony.com/

| Model | MAESTRO | Cerberus4 | GuitarSet | MusicNet | Slakh2100 | URMP |
|---|---|---|---|---|---|---|
| *Frame F1* | | | | | | |
| Hawthorne et al. (2021) | 0.66 | – | – | – | – | – |
| Manilow et al. (2020) | – | 0.63 | 0.54 | – | – | – |
| Cheuk et al. (2021) | – | – | – | 0.48 | – | – |
| Melodyne | 0.41 | 0.39 | 0.62 | 0.13 | 0.47 | 0.30 |
| MT3 (single dataset) | **0.88** | 0.85 | 0.82 | 0.60 | 0.78 | 0.49 |
| MT3 (mixture) | 0.86 | **0.87** | **0.89** | **0.68** | **0.79** | **0.83** |
| *Onset F1* | | | | | | |
| Hawthorne et al. (2021) | 0.96 | – | – | – | – | – |
| Manilow et al. (2020) | – | 0.67 | 0.16 | – | – | – |
| Cheuk et al. (2021) | – | – | – | 0.29 | – | – |
| Melodyne | 0.52 | 0.24 | 0.28 | 0.04 | 0.30 | 0.09 |
| MT3 (single dataset) | **0.96** | 0.89 | 0.83 | 0.39 | 0.76 | 0.40 |
| MT3 (mixture) | 0.95 | **0.92** | **0.90** | **0.50** | **0.76** | **0.77** |
| *Onset+Offset F1* | | | | | | |
| Hawthorne et al. (2021) | 0.84 | – | – | – | – | – |
| Manilow et al. (2020) | – | 0.37 | 0.08 | – | – | – |
| Cheuk et al. (2021) | – | – | – | 0.11 | – | – |
| Melodyne | 0.06 | 0.07 | 0.13 | 0.01 | 0.10 | 0.04 |
| MT3 (single dataset) | **0.84** | 0.76 | 0.65 | 0.21 | 0.57 | 0.16 |
| MT3 (mixture) | 0.80 | **0.80** | **0.78** | **0.33** | **0.57** | **0.58** |
| Mixture (Δ%) | -5.3 | +5.2 | +19.5 | +54.0 | +0.1 | +263 |

Table 2: Transcription F1 scores for Frame, Onset, and Onset+Offset metrics defined in Section 4.2. Across all metrics and all datasets, MT3 consistently outperforms the baseline systems we compare against. Dataset mixing during training ("mixture"), specifically, shows a large performance increase over single dataset training, especially for "low-resource" datasets like GuitarSet, MusicNet, and URMP. Percent increase over single-dataset training for Onset+Offset F1 is shown in the last row.

only use the output of the model's "guitar" head. (While Cerberus was not originally trained on GuitarSet, Manilow et al. (2020) uses GuitarSet as an evaluation dataset; we compare to Cerberus due to the lack of an alternative baseline for GuitarSet.) On MAESTRO and MusicNet, we compare to Hawthorne et al. (2021) and Cheuk et al. (2021), which were trained on those respective datasets.

Wherever possible, we provide results from baseline models computed on the same test split used to evaluate MT3. This may overestimate the performance of some baselines if tracks or track segments in our validation/test sets may have been included in the training sets of the baseline models (due to the lack of a consistent train/test/validation split for some of these datasets). In an effort to address this issue for future work, we provide exact details to reproduce our train/test/validation splits in Appendix B and give further details on the baselines used, including information on reproducing our results, in Section C. For each model, we compute the reported metrics using pretrained models provided by the original authors. The ability to provide evaluation results for a single model across several datasets is an additional benefit of our approach and a contribution of the current work.

## 4.3 RESULTS

Our main results are shown in Table 2, which compares our models' performance on the six datasets described above. Our model achieves transcription performance exceeding the current state of the art for each of the six datasets evaluated across all three standard transcription metrics (Frame, Onset, and Onset + Offset F1), as shown in Table 2. This is particularly notable due to the fact, mentioned above, that each baseline model was specifically designed (in terms of architecture and loss function), trained, and tuned on the individual datasets listed. Additionally, our model is able to significantly advance the state of the art on the three resource-limited datasets discussed above, GuitarSet, MusicNet, and URMP. Table 2 also demonstrates a large gain in performance on the

| MIDI Grouping | MAESTRO | Cerberus4 | GuitarSet | MusicNet | Slakh2100 | URMP |
|---|---|---|---|---|---|---|
| Flat | 0.81 | 0.74 | 0.78 | 0.33 | 0.48 | 0.62 |
| MIDI Class | 0.80 | 0.81 | 0.78 | 0.31 | 0.62 | 0.59 |
| Full | 0.82 | 0.76 | 0.78 | 0.34 | 0.55 | 0.50 |

Table 3: Multi-instrument F1 score for MT3 (mixture) trained and evaluated at different levels of instrument granularity. The **Flat** grouping treats all non-drum instruments as a single instrument. This resembles the setup of many prior "multi-instrument" transcription works which transcribe notes played by all instruments, without respect to their source. The **MIDI Class** grouping maps instruments to their MIDI class (Table 8). This creates groupings of eight program numbers each, with general classes for piano, guitar, bass, strings, brass, etc. The **Full** grouping retains instruments' program numbers as annotated in the source dataset. This requires the model to distinguish notes played by e.g. violin, viola, cello, all of which are grouped in the "strings" MIDI Class.

resource-limited datasets when using the mixture formulation of our task, particularly for the multi-instrument datasets MusicNet and URMP; the mixture performance leads to an Onset-Offset F1 gain of $54\%$ on MusicNet and $263\%$ on URMP. Our model outperforms other baselines specifically optimized for low-resource datasets, such as Cheuk et al. (2021), while also remaining competitive with or outperforming models tuned for large single-instrument datasets, i.e. Hawthorne et al. (2021).

Instruments "in the wild" come in many different forms, and labeling exactly which sound sources contain the same instrument is a necessary but nontrivial task. While the original General MIDI 1.0 specification (1996) provides a 1:1 mapping of program numbers to 128 instruments[4] and a coarser grouping of these instruments into "classes" of eight program numbers (Table 8), these mappings are necessarily reductive: not all instruments are represented in the original 128 instruments mapped to program numbers (e.g. ukulele), and other instruments (piano, organ, guitar) include multiple program numbers which may be desirable to treat as a single instrument for the purposes of transcription (e.g. transcribing program numbers 32-39 as a single "bass" class). To explore the effect of instrument label granularity, we train and evaluate models with three levels of instrument groupings: Flat, MIDI Class, and Full, shown in Table 3. We evaluate models at these three grouping levels according to the multi-instrument transcription metric defined in Section 4.2.

Table 3 demonstrates that our model makes few instrument label errors when it predicts onsets and offsets correctly, even at the highest level of granularity ("Full"), as the multi-instrument F1 scores are close to the onset-offset F1 scores in Table 2. We also provide an example transcription in Figure 3, which shows the distinct instrument tracks for an input from the Slakh2100 dataset. In addition to our transcription results, we provide further experimental results in Appendix D. There, we assess the ability of our approach to generalize to unseen datasets by conducting a set of leave-one-dataset-out (LODO) experiments; we also provide evidence regarding the label quality of our datasets by varying the onset and offset tolerance threshold used to compute F1 scores.

## 5  Conclusion and Future Work

In this work we have shown that posing multi-instrument music transcription as a sequence-to-sequence task and training a generic Transformer architecture simultaneously on a variety of datasets advances the state of the art in multi-instrument transcription, most notably in the low-resource scenario. We also introduced and applied a consistent evaluation method using note on-set+offset+instrument F1 scores, using a standard instrument taxonomy.

Our work suggests several future research directions. As labeled data for multi-instrument transcription with realistic audio is very expensive, transcription models may benefit from training on *un*labeled data, in a self- or semi-supervised fashion. There may also be value in a variety of data augmentation strategies, e.g. mixing together unrelated examples to generate new training data. Finally, high-quality AMT models such as ours present new frontiers for other musical modeling tasks, such as generative music modeling (i.e. Dhariwal et al. (2020); Huang et al. (2018)); the transcriptions from our model could be used as training data for a symbolic music generation model.

---

[4]https://en.wikipedia.org/wiki/General_MIDI#Program_change_events

## 6 REPRODUCIBILITY STATEMENT

In conjunction with the release of this work, we will make our model code, along with the code we used to replicate prior baseline models, available at `https://github.com/magenta/mt3`.

## 7 ETHICAL CONSIDERATIONS

One limitation of our system (and the baseline systems against which we compare) is that it is trained on and applicable only to music from the "Western tradition". The characteristic of Western music most relevant to this work is that it is typically composed of discrete notes belonging to one of 12 pitch classes i.e. "C", "C#", "D", etc. As such, music that does not have a well-defined mapping onto these 12 pitch classes is outside the scope of this work. This excludes many non-Western musical traditions such as Indian ragas and Arabic maqams, and Western genres like the blues that rely on microtonality. These types of music would be better suited to alternate representations e.g. single or multiple non-discretized pitch tracks. We note that such data should also be considered "low-resource" given the low availability of transcription datasets representing such traditions, and represent an important area for future work. See Holzapfel et al. (2019) for a user study on existing AMT systems in this context), and Viraraghavan et al. (2020) for an approach to transcription in one non-Western domain.

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

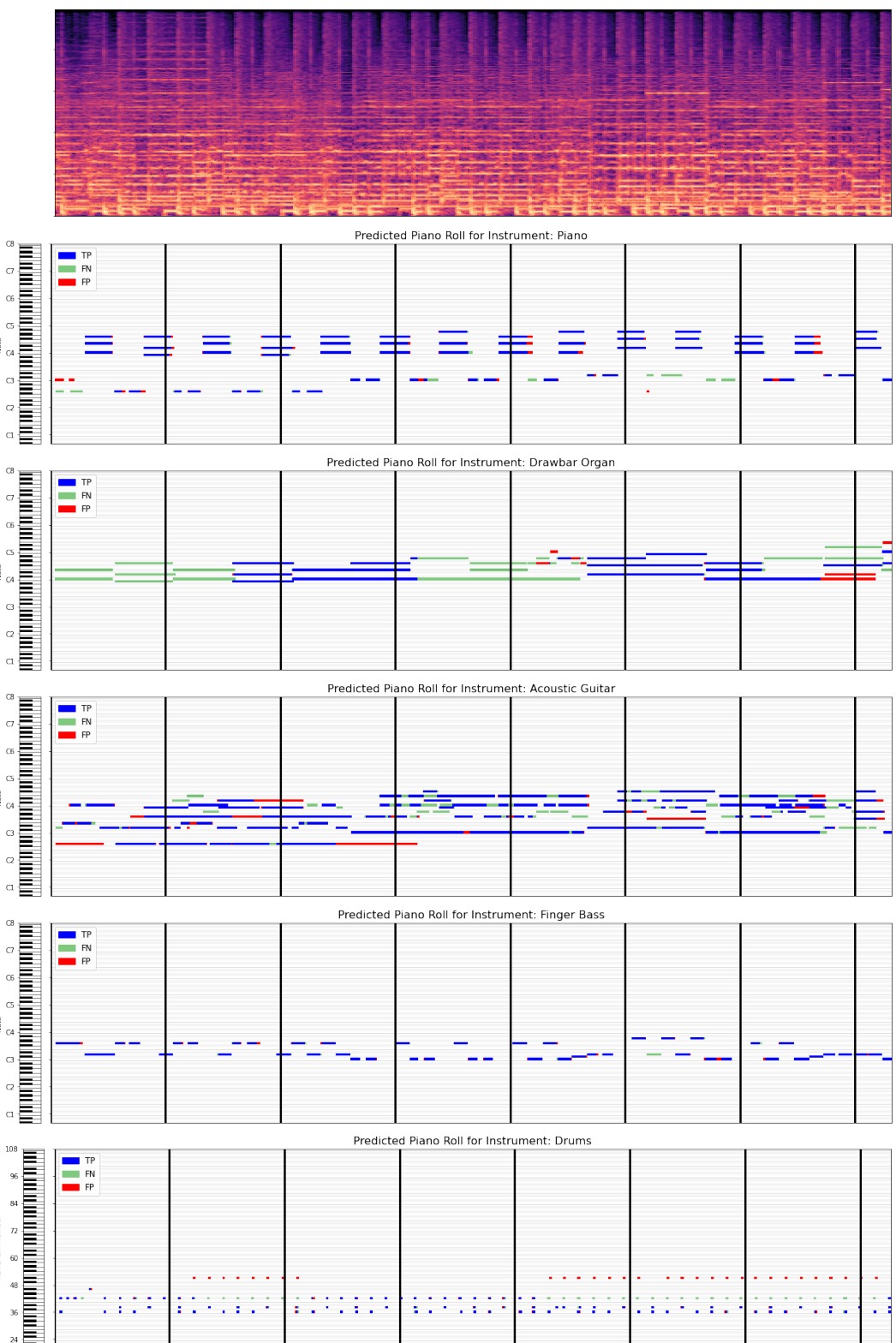

Figure 3: 15-second excerpt of MT3 transcriptions from a mix from the Slakh2100 dataset. Black lines indicate model input frames. Blue notes indicate "True Positive" notes with correct predicted onset, offset, pitch, and instrument. In this segment, the model achieves an Onset-Offset F1 of 0.665. More extensive results from MT3 can be found on the companion website at https://storage.googleapis.com/mt3/index.html.

| Model | MLP Dim | Num. Heads | Num. Layers | Embed. Dim. | LR | Params |
|-------|---------|------------|-------------|-------------|------|--------|
| T5 Small | 1024 | 6 | 8 | 512 | $1e-3$ | 93.7M |

Table 4: Details of model architecture used.

## A  MODEL AND TRAINING DETAILS

We use the T5 "small" model architecture described in Raffel et al. (2019), with the modifications defined in the T5.1.1 recipe[5]. This is a standard Transformer architecture, and we use the implementation available in `t5x`[6], which is built on FLAX (Heek et al., 2020) and JAX (Bradbury et al., 2020).

All mixture models are trained for 1M steps using a fixed learning rate of $0.001$. The dataset-specific models are trained for $2^{19}$ steps, as these models tended to converge much faster, particularly on the smaller datasets. Due to computational constraints we also train the LODO models for $2^{19}$ steps.

## B  DATASET DETAILS

This section describes the datasets used in the experiments throughout this work. Descriptive statistics for each dataset are provided in Table 1. However, because not all datasets provide an official train-test split, and because we perform preprocessing and filtering to extract suitable multi-instrument transcription datasets from the unprocessed versions of some datasets, we provide further details for reproducibility here.

### B.1  MAESTROV3

The MAESTRO (MIDI and Audio Edited for Synchronous TRacks and Organization) v3 dataset[7] (Hawthorne et al., 2018) contains 198.7 hours of piano performances captured via a Disklavier piano equipped with a MIDI capture device which ensures fine alignment ($\approx$3ms) between note labels and audio waveforms. The MAESTRO dataset contains mostly classical music and only includes piano performances (no other instruments).

MAESTRO includes a standard train/validation/test split, which ensures that the same composition does not appear in multiple subsets. 962 performances are in the train set, 137 are in the validation set, and 177 are in the test set. More detailed statistics on the MAESTRO dataset are available at `https://magenta.tensorflow.org/datasets/maestro`.

### B.2  SLAKH2100

The Lakh MIDI Dataset (Raffel, 2016) is a collection of 176,581 unique MIDI files scraped from publicly-available sources on the Internet, spanning multiple genres. The Synthesized Lakh Dataset (Slakh, or Slakh2100) (Manilow et al., 2019), is a dataset constructed by creating high-quality renderings of 2100 files from Lakh MIDI using professional-quality virtual instruments. The 2100 files selected all contain at least piano, bass, guitar, and drums, where each of these four instruments plays at least 50 notes.

When training on Slakh2100, we choose 10 random subsets of at least 4 instruments from each of the 2100 MIDI files as a form of data augmentation, expanding the number of training examples by a factor of 10 (though individual stems will in general appear in more than one example).

We use the standard Slakh2100 train/validation/test splits for all experiments.

---

[5]`https://github.com/google-research/text-to-text-transfer-transformer/blob/main/released_checkpoints.md#t511`
[6]`https://goo.gle/t5x`
[7]`https://magenta.tensorflow.org/datasets/maestro`

### B.3 CERBERUS4

We refer to as Cerberus4 another slice of the Slakh2100 dataset; in this case for each MIDI file we extract *all* subsets of instruments containing (exactly) one of each of piano, guitar, bass, and drums. This is intended to reflect the dataset construction in Manilow et al. (2020), but using entire tracks instead of shorter segments and with no additional criteria on instrument "activity".

Cerberus4 contains 1327 tracks representing 542.6 hours of audio. We use the Slakh2100 train/test/validation split to separate Cerberus4 tracks. The training set contains 960 tracks with 418.13 hours of audio, the test set contains 132 tracks with 46.1 hours of audio, and the validation set contains 235 tracks with 78.4 hours of audio.

### B.4 GUITARSET

GuitarSet[8] is a dataset consisting of high-quality guitar recordings and time-aligned annotations. GuitarSet contains 360 excerpts, which are the result of 6 guitarists each playing 30 lead sheets (songs) in two versions ("comping" and "soloing"). Those 30 lead sheets are a combination of five styles (Rock, Singer-Songwriter, Bossa Nova, Jazz, and Funk), three progressions (12 Bar Blues, Autumn Leaves, and Pachelbel Canon), and two tempi (slow and fast). The original GuitarSet annotations are provided in the JAMS format (Humphrey et al., 2014), which we convert to MIDI for use with standard evaluation libraries.

There is no official train-test split for GuitarSet. We establish the following split: for every style, we use the first two progressions for train and the final for validation. For convenience, we provide the exact train/validation split for tracks as part of the open-source release for this paper. This split produces 478 tracks for training, and 238 for validation.

### B.5 MUSICNET

MusicNet[9] (Thickstun et al., 2016) consists of 330 recordings of classical music with MIDI annotations. The annotations were aligned to recordings via dynamic time warping, and were then verified by trained musicians.

The standard train/test split for MusicNet Thickstun et al. (2016) only contains 10 test tracks and no validation set. We perform our own random split of the dataset into train/validation/test sets, and we provide the exact track IDs in each split in our open-source code release for this paper.

We discuss potential label quality issues with MusicNet in Appendix D.

### B.6 URMP

The University of Rochester Multi-Modal Music Performance (URMP) dataset (Li et al., 2018)[10] consists of audio, video, and MIDI annotation of multi-instrument musical pieces assembled from coordinated but separately recorded performances of individual tracks. That is, each part of each piece is recorded in isolation by an individual performer in coordination with the other performers (to ensure complete isolation of audio). The resulting mix is produced by combining the individual instrument tracks.

The dataset includes 11 duets, 12 trios, 14 quartets, and 7 quintets. In total, there are 14 different instruments in the dataset, including strings (violin, viola, cello, double bass), woodwinds (flute, oboe, clarinet, bassoon, soprano saxophone, tenor saxophone), and brass (trumpet, horn, trombone, tuba).

The dataset also includes videos and sheet music, which are not used in this paper.

We use the following pieces for validation: 1, 2, 12, 13, 24, 25, 31, 38, 39. The remaining pieces are used for training. This validation split reserves two duets; two trios; three quartets; and two quintets for the validation set, ensuring a diverse instrumentation in the validation split.

---

[8] https://GuitarSet.weebly.com/
[9] https://homes.cs.washington.edu/~thickstn/musicnet.html
[10] http://www2.ece.rochester.edu/projects/air/projects/URMP.html

We discuss potential label quality issues with URMP in Appendix D.

## C  BASELINE DETAILS

**Manilow et al. (2020):** For this baseline, we used a model trained with the authors' original code for the Cerberus 4-instrument model (guitar, piano, bass, drums) on the *slakh-redux* dataset, which omits duplicate tracks included in the original release of Slakh2100. We use the same procedure for randomly cropping audio and filtering for active instruments described above and in the original Cerberus paper (Manilow et al., 2020), using the public train/validation/test splits for Slakh.

**Cheuk et al. (2021):** We use the authors' pretrained models and inference script provided at `https://github.com/KinWaiCheuk/ReconVAT`. Due to resource limitations, following correspondence with the authors, we divide the audio tracks from MusicNet into 20-second segments for inference, and conduct evaluation on these segments directly. We discard any segments without any active notes in the ground-truth annotations, because the `mir_eval` metrics are undefined without any notes in the reference track.

**Melodyne:** Melodyne[11] is a professional-quality audio software tool designed to provide note-based analysis and editing of audio recordings. Melodyne contains multiple algorithms for polyphonic pitch tracking, including algorithms for "polyphonic sustain" (designed for instruments with a slow decay, such as strings) and "polyphonic decay" (designed for instruments with a fast decay. For all results in this paper, we used Melodyne Studio version 5.1.1.003. The raw .wav files for each dataset were imported into Melodyne, and the default pitch-tracking settings were used to transcribe the audio (which allows Melodyne to automatically select the algorithm most suited to a given audio file). Melodyne exports MIDI files directly, which were used for our downstream analysis.

Melodyne does not provide a programmatic interface. Due to the large amount of manual effort required to perform large-scale transcription with Melodyne, for MAESTRO, Slakh10, Cerberus4, and GuitarSet, our analysis of Melodyne is performed on a random subset of 30 tracks from the test set for each of these datasets. For URMP and MusicNet we evaluate Melodyne on the entire test set.

Because Melodyne is a proprietary third-party software tool, we are not available to provide further details on the exact algorithms used to transcribe audio.

## D  ADDITIONAL RESULTS

### D.1  EVALUATING ZERO-SHOT GENERALIZATION WITH LEAVE-ONE-DATASET-OUT

In order to evaluate the generalizability of our proposed model, we evaluate our model on a challenging zero-shot generalization task. The procedure in these 'leave-one-dataset-out' (LODO) experiments is as follows: Let the $\mathcal{D}_{\mathrm{tr},i}$ represent an individual transcription dataset (e.g. MAESTRO), such that the full training set is $\mathcal{D}_{\mathrm{tr}} \coloneqq \bigcup_i \mathcal{D}_{\mathrm{tr},i}$ For each dataset $\mathcal{D}_j$, we train an MT3 model using the same mixture procedure described above, but with training set $\tilde{\mathcal{D}}_{\mathrm{tr}} = \mathcal{D}_{\mathrm{tr}} \setminus \mathcal{D}_{\mathrm{tr},j}$. Then, we evaluate on each dataset $\mathcal{D}_{\mathrm{tr},i} \in \mathcal{D}_{\mathrm{tr}}$.

Since Slakh2100 and Cerberus4 are generated using the same subset of track stems and the same synthesis software, we jointly either include or exclude those datasets in our LODO experiments.

The results of this study are shown in Tables 5 and 6. Table 5 shows that our model is able to achieve nontrivial note prediction performance for most datasets, attaining multi-instrument F1 and onset-offset F1 scores which outperform the baseline models on each of the low-resource datasets (GuitarSet, URMP). For all of the datasets, our model achieves LODO onset F1 scores between 0.14 and 0.78. For the much more challenging multi-instrument F1 score, our model's performance on the LODO task varies by dataset.

In general, these results show that our model can obtain non-trivial transcription performance even for datasets it has never seen during training, despite large differences in the sonic qualities, compositional styles, and instrumentation of the zero-shot evaluation datasets. However, the LODO experiments also point to the sensitivity of the model to the absence of particular datasets, highlighting

---

[11]`https://www.celemony.com/en/melodyne/what-is-melodyne`

| Left-Out Dataset | Evaluation Dataset | | | | | |
|---|---|---|---|---|---|---|
| | **MAESTRO** | **Cerberus4** | **GuitarSet** | **MusicNet** | **Slakh2100** | **URMP** |
| *Frame F1* | | | | | | |
| **None** | 0.86 | **0.87** | **0.89** | 0.68 | **0.79** | **0.83** |
| **MAESTRO** | 0.60 | 0.86 | 0.89 | 0.69 | 0.75 | 0.82 |
| **Cerberus4 + Slakh2100** | **0.87** | 0.55 | 0.87 | 0.67 | 0.55 | 0.78 |
| **GuitarSet** | 0.86 | 0.86 | 0.58 | **0.71** | 0.76 | 0.82 |
| **MusicNet** | 0.86 | 0.86 | 0.89 | 0.53 | 0.76 | 0.79 |
| **URMP** | 0.86 | 0.86 | 0.89 | 0.71 | 0.76 | 0.76 |
| *Onset F1* | | | | | | |
| **None** | **0.95** | **0.92** | **0.90** | **0.50** | **0.76** | **0.77** |
| **MAESTRO** | 0.28 | 0.76 | 0.78 | 0.35 | 0.52 | 0.57 |
| **Cerberus4 + Slakh2100** | 0.82 | 0.21 | 0.75 | 0.30 | 0.14 | 0.49 |
| **GuitarSet** | 0.81 | 0.76 | 0.32 | 0.36 | 0.53 | 0.59 |
| **MusicNet** | 0.80 | 0.76 | 0.78 | 0.18 | 0.53 | 0.54 |
| **URMP** | 0.81 | 0.76 | 0.79 | 0.36 | 0.53 | 0.23 |
| *Onset+Offset+Program F1* | | | | | | |
| **None** | 0.80 | **0.80** | 0.78 | 0.33 | **0.57** | **0.58** |
| **MAESTRO** | 0.28 | 0.76 | 0.78 | 0.33 | 0.52 | 0.50 |
| **Cerberus4 + Slakh2100** | **0.82** | 0.07 | 0.75 | 0.29 | 0.02 | 0.42 |
| **GuitarSet** | 0.81 | 0.76 | 0.19 | **0.35** | 0.53 | 0.53 |
| **MusicNet** | 0.80 | 0.75 | 0.78 | 0.14 | 0.53 | 0.47 |
| **URMP** | 0.81 | 0.75 | **0.79** | 0.35 | 0.53 | 0.17 |

Table 5: Leave-one-dataset-out transcription scores.

| Training | Eval Dataset | | | | | |
|---|---|---|---|---|---|---|
| | **MAESTRO** | **Cerberus4** | **GuitarSet** | **MusicNet** | **Slakh2100** | **URMP** |
| *Frame F1* | | | | | | |
| **Full mixture** | 0.86 | 0.87 | 0.89 | 0.68 | 0.79 | 0.83 |
| **Zero-shot** | 0.60 | 0.55 | 0.58 | 0.53 | 0.55 | 0.76 |
| *Onset F1* | | | | | | |
| **Full mixture** | 0.95 | 0.92 | 0.90 | 0.50 | 0.76 | 0.77 |
| **Zero-shot** | 0.28 | 0.21 | 0.78 | 0.18 | 0.14 | 0.23 |
| *Onset+Offset+Program F1* | | | | | | |
| **Full mixture** | 0.80 | 0.80 | 0.78 | 0.33 | 0.57 | 0.58 |
| **Zero-shot** | 0.28 | 0.07 | 0.19 | 0.14 | 0.02 | 0.17 |

Table 6: Zero-shot transcription scores.

the resource-constrained nature of the available music transcription datasets even when combined. For example, the Slakh2100 + Cerberus4 combination is the only dataset in our LODO experiments that contains bass and synthesizer; without training on those datasets, the model is unable to learn to identify these instruments.

### D.2 ONSET-OFFSET THRESHOLD SENSITIVITY ANALYSIS

There are many reasons that one transcription dataset may be more difficult than another. In the course of our experiments, we observed potential errors in labeling for some of our datasets, including incorrect onset/offset times, particularly in MusicNet and URMP. The original MusicNet paper estimated an error rate of around 4% Thickstun et al. (2016); the error rate of URMP annotations has not been investigated, to our knowledge.

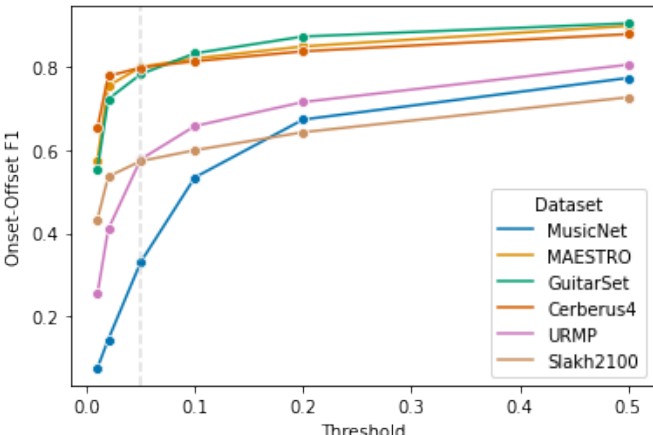

Figure 4: Onset-Offset F1 performance of our model over varying thresholds for the Onset-Offset F1 metric described in Section 4.2. The default threshold of 50 ms is indicated by a dashed line.

While a direct investigation of labeling errors is beyond the scope of this work, we present some initial evidence regarding label *timing* errors in Figure 4. Here, we systematically increase the tolerance $t$ used for the Onset-Offset F1 metric (described in Section 4.2) over a grid of values for $t \in [10\text{ms}, 500\text{ms}]$, and compute the Onset-Offset F1 for MT3 using threshold $t$ for both onset and offset. Our results are consistent with the presence of label timing errors in both URMP and MusicNet: As the threshold increases, performance on datasets with high-quality timing labels tends to level off to a baseline value. However, performance on URMP and MusicNet continues to increase as the threshold increases, which is suggestive of large timing errors beyond the standard 50 ms threshold used to evaluate the performance of all models in the experiments in our work.

These results suggest that MusicNet and also potentially URMP may be affected by label timing issues which could affect the learning, quality, and generalizability of models trained on these datasets (particularly considering that the default threshold for a correct prediction is only 50 ms, small timing errors can significantly increase the difficulty of properly modeling a dataset with noisy timing labels). While there is visible evidence of labeling errors[12] in the MusicNet dataset upon inspection, i.e. using the MusicNet inspector tool[13], we are not aware of scholarly work which has formally investigated this important issue to date. We encourage further investigation into labeling issues on these datasets.

## E  MIDI CLASS GROUPINGS

Program numbers for the "MIDI Class" grouping results described in Table 3 match the original MIDI classes from the original specification, where MIDI group for program $p$ corresponds to floor $(p/8)$; we give a complete listing of the MIDI program number to MIDI class mappings in Table 8. However, for the Cerberus4 and SLAKH2100 datasets, instruments are grouped by "class", which is a categorization of groups of patches used to synthesize those instruments. We construct a mapping of SLAKH "class" to MIDI program numbers, given in Table 7. These program numbers are applied to each "class" in the Cerberus4 and SLAKH2100 datasets as a simple lookup, and the associated program numbers are used.

## F  OPEN-SOURCE IMAGE ATTRIBUTION

The instrument icons used in Figure 1 are used under the Creative Commons license via the Noun Project. We gratefully acknowledge the following creators of these images:

---

[12]http://disq.us/p/236ypfr
[13]https://musicnet-inspector.github.io/

| Slakh2100 Class | MT3 Instrument Token Number |
|---|---|
| Acoustic Piano | 0 |
| Electric Piano | 4 |
| Chromatic Percussion | 8 |
| Organ | 16 |
| Acoustic Guitar | 24 |
| Clean Electric Guitar | 26 |
| Distorted Electric Guitar | 29 |
| Acoustic Bass | 32 |
| Electric Bass | 33 |
| Violin | 40 |
| Viola | 41 |
| Cello | 42 |
| Contrabass | 43 |
| Orchestral Harp | 46 |
| Timpani | 47 |
| String Ensemble | 48 |
| Synth Strings | 50 |
| Choir and Voice | 52 |
| Orchestral Hit | 55 |
| Trumpet | 56 |
| Trombone | 57 |
| Tuba | 58 |
| French Horn | 60 |
| Brass Section | 61 |
| Soprano/Alto Sax | 64 |
| Tenor Sax | 66 |
| Baritone Sax | 67 |
| Oboe | 68 |
| English Horn | 69 |
| Bassoon | 70 |
| Clarinet | 71 |
| Pipe | 73 |
| Synth Lead | 80 |
| Synth Pad | 88 |

Table 7: Mapping of Slakh2100 "classes" to MT3 Instrument Token numbers used for all experiments using the Slakh2100 dataset (i.e., all columns labeled Slakh2100 in experiments throughout this paper). Slakh2100 classes have slightly more granularity than the 16 MIDI Classes (see Table 8 and our MT3 Instrument tokens are designed to roughly correspond MIDI program numbers.

| MIDI Program Numbers | Instruments |
|:---:|:---|
| *Piano* | |
| 1-8 | Acoustic Grand Piano, Bright Acoustic Piano, Electric Grand Piano, Honky-tonk Piano, Electric Piano 1, Electric Piano 2, Harpsichord, Clavinet |
| *Chromatic Percussion* | |
| 9-16 | Celesta, Glockenspiel, Music Box, Vibraphone, Marimba, Xylophone, Tubular Bells, Dulcimer |
| *Organ* | |
| 17-24 | Drawbar Organ, Percussive Organ, Rock Organ, Church Organ, Reed Organ, Accordion, Harmonica, Tango Accordion |
| *Guitar* | |
| 25-32 | Acoustic Guitar (nylon), Acoustic Guitar (steel), Electric Guitar (jazz), Electric Guitar (clean), Electric Guitar (muted), Electric Guitar (overdriven), Electric Guitar (distortion), Electric Guitar (harmonics) |
| *Bass* | |
| 33-40 | Acoustic Bass, Electric Bass (finger), Electric Bass (picked) , Fretless Bass, Slap Bass 1, Slap Bass 2, Synth Bass 1, Synth Bass 2 |
| *Strings* | |
| 41-48 | Violin, Viola, Cello, Contrabass, Tremolo Strings, Pizzicato Strings, Orchestral Harp, Timpani |
| *Ensemble* | |
| 49-56 | String Ensemble 1, String Ensemble 2, Synth Strings 1, Synth Strings 2, Choir Aahs, Voice Oohs, Synth Voice or Solo Vox, Orchestra Hit |
| *Brass* | |
| 57-64 | Trumpet, Trombone, Tuba, Muted Trumpet, French Horn, Brass Section, Synth Brass 1, Synth Brass 2 |
| *Reed* | |
| 65-72 | Soprano Sax, Alto Sax, Tenor Sax, Baritone Sax, Oboe, English Horn, Bassoon, Clarinet |
| *Pipe* | |
| 73-80 | Piccolo, Flute, Recorder, Pan Flute, Blown bottle, Shakuhachi, Whistle, Ocarina |
| *Synth Lead* | |
| 81-88 | Lead 1 (square), Lead 2 (sawtooth), Lead 3 (calliope) , Lead 4 (chiff), Lead 5 (charang), Lead 6 (space voice), Lead 7 (fifths), Lead 8 (bass and lead) |
| *Synth Pad* | |
| 89-96 | Pad 1 (new age or fantasia), Pad 2 (warm), Pad 3 (polysynth or poly, Pad 4 (choir), Pad 5 (bowed glass or bowed), Pad 6 (metallic), Pad 7 (halo), Pad 8 (sweep) |
| *Synth Effects* | |
| 97-111 | FX 1 (rain), FX 2 (soundtrack), FX 3 (crystal), FX 4 (atmosphere), FX 5 (brightness), FX 6 (goblins), FX 7 (echoes or echo drops), FX 8 (sci-fi or star theme) |
| *Other* | |
| 105-112 | Sitar, Banjo, Shamisen, Koto, Kalimba, Bag pipe, Fiddle, Shanai |
| *Percussive* | |
| 113-120 | Tinkle Bell, Agogô, Steel Drums, Woodblock, Taiko Drum, Melodic Tom or 808 Toms, Synth Drum, Reverse Cymbal |
| *Sound Effects* | |
| 121-128 | Guitar Fret Noise, Breath Noise, Seashore, Bird Tweet, Telephone Ring, Helicopter, Applause, Gunshot |

Table 8: All instruments as defined by the MIDI specification Association (1996), grouped by MIDI Class (rows with grey background), program numbers, and their associated instrument names.

- Piano by Juan Pablo Bravo from the Noun Project.
- Guitar by varvarvarvarra from the Noun Project.
- Bass by Josue Calle from the Noun Project.
- Drum Set by Sumyati from the Noun Project.
- Oboe by Rank Sol from the Noun Project.
- Clarinet by Pham Thanh Lôc from the Noun Project.
- French Horn by Creative Stall from the Noun Project.
- Bassoon by Lars Meiertoberens from the Noun Project.
- Flute by Symbolon from the Noun Project.
- Violin by Benedikt Dietrich from the Noun Project.
- (Electric) Piano by b farias from the Noun Project.
- Viola by Vasily Gedzun from the Noun Project.
- Violin by Francesco Cesqo Stefanini from the Noun Project.
- Cello by Valter Bispo from the Noun Project.
- (String) bass by Soremba from the Noun Project.
- Acoustic guitar by farra nugraha from the Noun Project.

