# OpenReview forum: "MT3: Multi-Task Multitrack Music Transcription"
_ICLR.cc/2022/Conference — ICLR 2022 Spotlight_

### Official Review · Reviewer_RijS · 2021-11-01

**Correctness:** 4
**Technical Novelty And Significance:** 4
**Empirical Novelty And Significance:** 4
**Recommendation:** 8
**Confidence:** 4

**Main Review:**

- Should we really call it multi-task? I'm not sold that the difference in target instruments is significant enough to call it multi-task.
- Experiment > Frame F1 - is there any reason for the choice of 62.5 frame per second?
- I might have missed this from the text - How exactly were the single-instrument metrics computed? Considering every note a single instrument? If it's not specified in the paper, please add it.
- It's great that even Melodyne was included in the experiment.
- Re: Table 3 - please add the grouping strategies in the appendix, or at least at the demo page. I expect this work will be impactful in the field, and as a consequence, the grouping strategies would need to be shared somewhere.
- Appendix A seems to be a bit too simple to me. T1 "small" model - does it really define everything about the model?


**Summary Of The Paper:**

The paper introduces a system called MT3 that performs music transcription with various types of datasets. In the experiment, the system showed state-of-the-art performance in all the relevant metrics and on the selected datasets, which are comprehensive.

**Summary Of The Review:**

Very solid work. A few more details about the experiment and the model would be great.

---

> ### Author Response · Authors · 2021-11-20
> **Author Response to Reviewer RijS**
>
> We are grateful to reviewer 4 for their thorough evaluation, and hope that, as the reviewer states, our work will be “impactful in the field.”
>
> We agree with the reviewer that we are using the term “multi-task” in a new manner for this paper, but would like to further motivate our reasoning for stretching our license to do so. The main motivation for differentiating between “multi-task” and “multi-instrument/multitrack” is to highlight the contribution of this work using a single network to train on multiple datasets (“tasks”) that previously required individually-tailored networks for each dataset. Previous work has trained a single network for multi-instrument transcription within a dataset, but fails to generalize to training jointly on multiple datasets with arbitrary combinations of instruments (See “Onsets and Frames by [3], or the Cerberus model by [2]). We believe this framing is in line with, and intended to draw parallels to, comparable work in NLP. “Multi-task” models such as the T5, from which our model is inherited, are called such because they have the ability to train a single model on multiple datasets that previously would have required domain-specific network adaptations for each dataset [1]. We understand the perspective of multi-task referring to semantically different functions (e.g. classification, segmentation, regression), but as described above felt that highlighting the key insight of the paper (applying NLP dataset-agnostic training to Music Information Retrieval) hopefully merited the flexibility with the term.
>
> The single-instrument metrics were computed by considering each instrument as a distinct “track”, and then computing the metric over each track; we aggregate these scores in the typical method for F1 aggregation. Our implementation of the metric will also be available in our open-source code release. Additionally, as the reviewer requested further details about the model, we do believe that “t5 small” encapsulates all of the architectural details needed for i.e. reproducing the model; see [1] Sec. 3.6-3.7. This is an explicit configuration available within the open-source T5 codebase (see https://github.com/google-research/text-to-text-transfer-transformer/blob/main/t5/models/gin/models/t5.1.1.small.gin and associated files). Additionally, please note that the open-source release of our code will make available the complete model specification and training code, along with all architectural and training hyperparameters.
>
> Regarding the reviewer’s clarification, we elected to use 62.5 frames per second based on the heuristic that this aligns to 256 samples per frame at our audio sampling rate, and also cleanly divides the FFT hop size used to compute the spectrograms. There does not appear to be an agreed-upon standard in the field for this value, and we hoped to at least transparently report our own for future comparisons to our work. Additionally, we evaluated other values for this threshold and found that the metrics, both for MT3 and the baseline models, did not show much sensitivity to moderate changes in the frames per second.
>
> The grouping strategy we used was the standard MIDI class grouping, which is floor(program_number / 8) and is described in the MIDI specification, along with the Wikipedia table linked in Footnote 4. However, we agree that explicitly providing this grouping strategy may be useful to readers; we added new tables in the Appendix (Table 8) with the explicit program numbers and instrument names associated with each MIDI class. Another potential ambiguity is the mapping of Slakh “classes” (a concept introduced in [2]) to MIDI programs; we provide a new table with this mapping as well (Table 7).
>
> References:
>
> [1] Raffel, Colin, et al. "Exploring the Limits of Transfer Learning with a Unified Text-to-Text Transformer." Journal of Machine Learning Research 21 (2020): 1-67.
>
> [2] Manilow, Ethan, et al. "Cutting music source separation some Slakh: A dataset to study the impact of training data quality and quantity." 2019 IEEE Workshop on Applications of Signal Processing to Audio and Acoustics (WASPAA). IEEE, 2019.
>
> [3] Hawthorne, Curtis, et al. "Onsets and frames: Dual-objective piano transcription." arXiv preprint arXiv:1710.11153 (2017).

---

### Official Review · Reviewer_zmLf · 2021-11-02

**Correctness:** 4
**Technical Novelty And Significance:** 2
**Empirical Novelty And Significance:** 4
**Recommendation:** 8
**Confidence:** 5

**Main Review:**

This paper seems to be a great milestone in the AMT research. It is probably the first unified AMT model that can take music audio with an arbitrary number of instruments. Most of all, leveraging the ability of being trained with all types of AMT datasets, it achieves impressive improvement in the accuracy.

< Good parts >
- Drum is always a special track in MIDI because the same contains different drum samples over different MIDI notes.  It is great to have drum tokens explicitly added.
- The introduction of "end of tie section" token is a nice idea, considering the nature of music data which requires input audio to be segmented with a fixed length in the practical experiment setting.
- The evaluation of instrument labeling is conducted as three different granularity of instrument grouping. In particularly, MIDI class, based on the instrument family, make a great sense.
- The generalization test to out-of-domain data with a series of leave-one-dataset-out experiments is also valuable. It emphasizes the strength of the proposed model and well-aligned with the universality.
- It is interesting to see the performance of the commercial software (Melodyne, which is known to be based on DSP) in comparison to the previous models.

< Weak parts >
- The main contribution is the MIDI-like data representation for the output. This representation was "hand-designed" rather than learned. I understand that it is natural to design the output form manually and can be seen as a case of knowledge integration to learning models. However, the output representation was simply applied to an existing Transformer model (T5 small) with minor change. In other words, technical novelty is weak.
- The proposed model was not evaluated with datasets with vocal which is the most essential sound source in popular music. This might be due to the lack of datasets which include vocal tracks in mixtures and their labels. However, the lack of vocal track evaluation limits the capacity of the model.  Since vocal is one of the most expressive instruments, it would be interesting to see how the model works for vocal with background instrumental sounds.

< Questions / Minor parts >
- The demo examples show that the input audio and output MIDI are temporally aligned well. I wonder if this is because of the absolute positional encoding. What happen if the relative positional encoding?
- page 7: "...if it is has the same pitch..." -->  "if it has the same pitch"


**Summary Of The Paper:**

This paper presents a universal automatic music transcription (AMT) model agnostic to the number of instrument tracks. The model is trained across various AMT datasets, each of which was used as a different AMT task, and less biased to the volume of individual datasets. This is enabled by the novel MIDI-like output data representation designed to include "instrument" tokens and also supplementary tokens such as "tie" or "EOS" tokens. The authors conducted comprehensive experiments with 6 datasets, calculated not only regular accuracy metrics (Frame F1, Onset F1, Onset-Offset F1) but also multi-instrument F1 which is a new metric for the universal AMT task. They significantly improved the accuracy over all datasets and also showed that the model trained with mixed datasets are generally more effective than those trained with a single dataset.

**Summary Of The Review:**

This paper achieved a great milestone in AMT. The demo examples are very impressive and show great potential as a musical tool or a data generator for symbolic music modeling. The main weakness is that the main contribution is designed by hands, in other words, the technical novelty is low. Also, vocal tracks are not included in the experiment. But, this seems to be a matter of dataset availability.

---

> ### Author Response · Authors · 2021-11-20
> **Author Response to Reviewer zmLf**
>
> We are grateful to the reviewer for their insightful comments. We are delighted that the reviewer shares our opinion that this work may be “a great milestone in AMT research”, and providing such a model was a goal of this work.
>
>  The reviewer is correct that our work uses an off-the-shelf T5 model with only small changes; however, we feel that this is an advantage of our approach. While the architecture may not be novel, our main contribution is in framing multi-instrument music transcription as a sequence-to-sequence task, and empirically advancing the state of the art in this area as a result. We believe that the simple architecture, in contrast to the complex, handcrafted architectures and loss functions of prior work, are an advantage of our approach. Similarly, while we agree that a fully “learned” representation for this task would be enticing, it is not clear how to develop such a representation; we hope that our general output vocabulary is sufficiently flexible to handle a wide variety of musical representations (and is also extensible through the addition of arbitrary new tokens to represent additional musical events).
>
> The reviewer is correct that our work does not evaluate audio with vocals, which we agree are an important aspect of transcription, and that scarcity of open-sourced datasets with permissive licensing was a factor in this limitation. Nevertheless, we believe that our current model does demonstrate an important step toward modeling a variety of musical sources. We plan to explore adding vocal transcription capabilities in the future. Additionally,we hope that the open-source release of our model and code will allow for the research community to investigate MT3’s performance on vocal performances, and to explore further extensions to vocal datasets.
>
> For results on the use of relative time shifts, we refer the reviewer to (Hawthorne et al., Sequence-to-Sequence Piano Transcription with Transformers, ISMIR’21), Table 1, which shows that relative time shifts significantly degraded the model performance. Due to this degradation, we elected not to explore relative time shifts in the multi-instrument model.

---

### Official Review · Reviewer_vgm4 · 2021-11-02

**Correctness:** 3
**Technical Novelty And Significance:** 3
**Empirical Novelty And Significance:** 3
**Recommendation:** 8
**Confidence:** 3

**Main Review:**

The contribution of the paper is mainly written in the "Summary of the paper" part, I will write some questions in this section.

I think multi-instrument music transcription task can be regarded as two parts which are transcription (addressing note) and classification (instrument). As the performance of the transcription part can be enhanced apart from the instrument classification part when a large transcription data is used for training, I wonder the performance of a model trained without instrument supervision (without instrument parts in token). This may give some insights about the effect of the multi-instrument token.

Since the main target was on dataset-wise experiment to measure zero-shot performance, the main results are all reported in dataset-wise. However, I wonder the performance difference between instrument-wise split. For example, piano, bass, guitar, ... split (no matter each instrument contains mixed dataset). I think this will give some insights on instrument-level analysis.


**Summary Of The Paper:**

This paper proposes a multi-task multi-track music transcription framework. Music transcription task has mainly been tackled individually for each instrument type. However, in this work, the authors jointly trained the model using several datasets with different instrument types. As a result, the proposed model provides better transcriptional results compared to models trained on individual data sets. The contribution of the paper follows three parts. 1. multi-task transcription model: This is a good direction of tackling low-resourced transcription task. 2. MIDI-like representation for learning multi-instrument piano rolls: Similar ideas are explored in MIDI-based music generation work recently, and it is good to see this direction is also proposed in multi-instrument transcription task. 3. multi-instrument F1 score metric: Since multi-instrument transcription task is somewhat new, having this kind of metric will be a good for following researches.


**Summary Of The Review:**

Overall, the paper is well-written, and I vote for accepting the paper. The authors proposed dataset split, model, and some evaluation metric for somewhat new task.

---

> ### Author Response · Authors · 2021-11-20
> **Author Response to Reviewer vgm4**
>
> We are grateful to the reviewer for their thoughtful assessment of our work.
>
> The reviewer raises an important question about per-instrument performance of our model. As the reviewer is aware, instrument categorization is nontrivial, and we evaluate three different groupings (Flat, MIDI Class, Full) when computing multi-instrument F1. As such, any of these categorizations could be used for reporting various flavors of instrument-wise performance; for example, reporting by “MIDI Class” would result in 16 “instruments”, and reporting by “Full” would result in 128 (however, a small number of instruments never occur in the training set when using the full list of 128 MIDI program numbers); we also recognize that a highly non-uniform distribution of instruments, and the correlations between instruments which tend to co-appear in various contexts, would potentially bias these results and complicate their interpretation. While we do not report instrument-specific metrics in our work, we did add clarification on these groupings in a new supplementary section entitled “MIDI Class Groupings”. Additionally, we hope that the open-source release of our code and model will support future, more detailed investigations of the various performance dimensions of MT3.
>
> The reviewer asks about the performance of a model trained without instrument supervision: we note that “flat” MIDI grouping (equivalently, onset-offset F1) gives the result of transcription performance without respect to instrument, and so the first row of Table 3 gives these results.

---

### Official Review · Reviewer_4Man · 2021-11-02

**Correctness:** 3
**Technical Novelty And Significance:** 3
**Empirical Novelty And Significance:** 4
**Recommendation:** 8
**Confidence:** 4

**Main Review:**

Strong points & contributions:

- Combining AMT datasets to provide a unified training framework
- Outperforming the relevant state-of-the-art in AMT
- Widening the scope into multi-instrument transcription
- Systematic analysis of model training under diverse setups
- Comparison of numerous architectures against numerous evaluation metrics.
- A novel, musically relevant evaluation metric taking instruments into consideration
- Out-of-dataset transcription experiments

Below I suggest minor improvements and future work:

- An ideal AMT system should be capable of transcribing multiple instruments at once.

  I think the notion of "ideal" is ill-defined. Any music transcription, let it be human or machine annotated, and carried for an engineering or musicological task in mind, should be tailored for the analytic purpose. The purpose does not necessarily encompass robust handling of multiple instruments, time/pitch precision, or else. For example, the most crucial goal may be obtaining a music score like explained in:

  `Carvalho R. G. C., Smaragdis P. (2017). Towards end-to-end polyphonic music transcription: transforming music audio directly to a score. In 2017 IEEE Workshop on Applications of Signal Processing to Audio and Acoustics (WASPAA), New Paltz, NY, USA, pp. 151–55.`

and if the method's output notation is not satisfactory for human users, the results will be far from desired.

- The authors clearly demonstrate the case of AMT having relatively low resources. However, they omit to discuss the (even lower resourced) studies applied to music out of 12 tone-equal-temperament (see suggestions below). The authors should include at least the recent work such as:

  `Holzapfel A., Benetos E. (2019). Automatic music transcription and ethnomusicology: a user study. In Proceedings of the 20th International Society for Music Information Retrieval Conference (ISMIR), Delft, Netherlands, pp. 678–84.`

  `V. S. Viraraghavan, A. Pal, H. Murthy, and R. Aravind, "State-Based Transcription of Components of Carnatic Music," ICASSP 2020 - 2020 IEEE International Conference on Acoustics, Speech, and Signal Processing (ICASSP), 2020, pp. 811-815, DOI: 10.1109/ICASSP40776.2020.9054435.`

- The authors should contrast their metrics of choice and the novel metric in the paper with the MV2H metric proposed in:

  `McLeod A., Steedman M. (2018). Evaluating automatic polyphonic music transcription. In Proceedings of the 19th International Society for Music Information Retrieval Conference (ISMIR), Paris, France, pp. 42–49.`

- Figure 2: The color selection is not print or (more importantly) color-blind friendly. I would suggest the authors re-render the Figures. See: https://www.nature.com/articles/nmeth.1618 for a reference

- Section 3.2: It would have been better to introduce the explanation in the order of the tokens in Figure 2 for readability.

- On/Off ... events are interpreted as note-on or note-off.

  It might be better to explain what on-off means (e.g., a note is played/released) for an audience unfamiliar with MIDI specifications.

- audio is split into smaller, non-overlapping segments ...

  Apart from the "turn-off" handled by the "end of tie token," does the model performance degrade around the edges of the segment?

- 4. Experiments, ... labeling issues with certain datasets

  A few other dimensions could be:

  - different analytic purposes
  - difference between the granularity of the transcriptions across the datasets
  - transcriber consistency/reliability

  It could be helpful to exemplify such issues -at least qualitatively- in the appendix.

- Section C, Section D, ...

  I think they should be Appendix C, Appendix D.

- Ethical considerations

  I think the text in this section does not describe ethical considerations but technical constraints.
  Nit: See http://globalnotation.org.uk/ as another alternate representation for encoding micro/finer-grained music.

Suggestions for future work:
=====

- entirely out-of-domain

  A nitpick to rather suggest future work: I'm afraid I have to disagree that leaving each dataset out is an **entirely** out-of-domain setting; the datasets share common instruments, genres, rhythmic structure, temperament, etc. Having said that it would be interesting to observe how MT3 behaves as the test data gradually becomes out-of-domain, e.g., instrumental vs voice, inserting compound rhythms, altering the timbre/instruments, zero-shot transcription on a LakhNES-like dataset or traditional music datasets (Meertens Tune Collection, Greek folk tunes in Benetos & Holzapfel's recent publications, CompMusic corpora ...) in the future.

- As the authors argue, the characteristics and the annotation alignment varies between the datasets. The work may open an exciting path towards studying "data valuation" for automatic music transcription, e.g., which parts of the data are more informative, inconsistent, or erroneous. ds3labs's (https://ds3lab.inf.ethz.ch/easeml.html) work may be of inspiration.

**Summary Of The Paper:**

The authors combine numerous automatic music transcription datasets to devise a training framework. They train an off-the-shelf T5 architecture on the combined data and outperform the current state-of-the-art reported on individual datasets. They report numerous experiments to demonstrate the robustness of the model on low resource datasets & out-of-dataset transcription against task-informed variations of F1-scores, including a novel multi-instrument-based F1-score introduced in the paper.



**Summary Of The Review:**

The paper is well written. The literature review is extensive - even though there can be several additions (see the specific comments in the main review). The work extends the scope of existing work in AMT significantly. The experiments are thorough and the authors declare that they will present the necessary code, experimental setup, and results in the camera-ready version so that the work may be reproducible.

Given the work's strong points as described above, I would like to recommend the paper for publication in ICLR 2022.

---

> ### Author Response · Authors · 2021-11-20
> **Author Response to Reviewer 4Man**
>
> We are grateful to the reviewer for their thoughtful suggestions.
>
> We added the suggested citations to our paper. In particular, we added a note comparing our metric to the MV2H metric of (Mcleod and Steedman); we note that multi-instrument F1 is stricter than MV2H (as MV2H ignores offsets, and eliminates notes from ground-truth during evaluation when the pitch is not correctly detected). We also believe that multi-instrument F1 allows for more straightforward comparison to prior work, which used (non-multi instrument) variants of F1 with the same underlying implementation from the mir_eval library. We updated Fig. 2 to the colorblind-friendly palette in the suggested citation; we thank the reviewer for helping to make our work more accessible. We also made the adjustments for clarity as the reviewer suggests, including attempting to emphasize that our model outputs MIDI-like tokens and not e.g. a musical score, which could be another useful output representation.
>
> The reviewer asks whether model performance degrades around the segment boundaries. As the reviewer notes, our “ties” section addresses the main issue we observed related to the input chunking – as the paper describes (Section 3.2), sometimes the model would forget to “turn off” notes that had been playing in a previous segment. Forcing the model to declare whether there were “ties” carried over from the previous segment (and turning notes off otherwise) helped to significantly reduce this behavior. Otherwise, we notice that the model sometimes fails to “carry over” instrument labels, which manifests as instruments occasionally changing within a phrase over time (i.e., the first 10s of the final example of the “in-the-wild” section in the online supplement). Additionally, we also would direct the reviewer to the many other examples on our online supplement, which generally do not show visible/audible artifacts due to segment boundaries (which would occur every ~2s in the input audio).
>
> We understand the reviewer’s point about “ethical considerations” vs. “technical limitations”. We titled the section as such to ensure it was clearly our paper’s “ethics statement” as described by the ICLR 2022 author guide, but agree there is strong overlap between technical limitations and their ethical implications. Nonetheless, we did wish to highlight that certain styles of music are excluded from our transcription setup.  We also added a reference to the “global notation” project the reviewer helpfully pointed out. We hope that the open-source release of code associated with our work will potentially support future research using alternative representations of music; an MT3 model with a modified vocabulary could seemingly support transcription of several alternative (non-MIDI-like) representations.
>
> The reviewer also pointed out useful directions for testing more “out-of-domain” examples (of course, and we understand their point that many forms of music share some high-level structures and may be considered as similar domains). We have conducted analyses along the lines of the reviewers’ suggestion, transcribing “in-the-wild” examples drawn from a variety of audio sources not in the training set. We have added examples of these to our supplementary page (under the heading “in-the-wild examples”; see https://storage.googleapis.com/mt3/index.html#in-the-wild) but look forward to performing more such transcriptions. We also expect that users will begin creating their own for a variety of domains via an open online tool we plan to offer which will enable transcriptions of user-uploaded audio via an interactive Python notebook. We softened our previous language regarding “entirely out-of-domain” transcription in the leave-one-dataset-out experiments.

---

> > ### Comment · Reviewer_4Man · 2021-11-20
> > **Thanks for the responses and changes**
> >
> > I would like to thank the authors for their responses and rigor. I would like to congratulate them another time for this work. Hopefully, it will be pivotal for AMT research for the next couple of years.
> >
> > I went through the corrections and I am happy with the changes and responses. In particular:
> >
> > - I agree on MV2H being less strict and it's good that the paper explicitly compares with the past work.
> > - The "in-the-wild" examples are spectacular!
> > - I hope that researchers will pick up on your excellent work to further generalize the framework beyond 12-TET.
> >
> > While checking the corrections I noticed two minor editorial issues:
> >
> > - Section C, Section D, ... => I think they should be Appendix C, Appendix D. This is from my original review. If I'm mistaken please ignore this comment
> > - There are some inconsistencies in the references, e.g. "Proc. vs Proceedings vs In conference" or how ICASSP is displayed. Please make sure that the references are consistent with each other.

---

### Decision · Program_Chairs · 2022-01-20

**Decision:**

Accept (Spotlight)

**Comment:**

This work concerns Automatic Music Transcription (AMT) -- transcribing notes given the audio of the music. The paper demonstrates that a single general-purpose transformer model can perform AMT for many instruments across several different transcription datasets. The method represents the first unified AMT model that can transcribe music audio with an arbitrary number of instruments.

All reviewers rated this paper highly and are excited about seeing it at the conference. One reviewer noted that "This paper seems to be a great milestone in the AMT research. It is probably the first unified AMT model that can take music audio with an arbitrary number of instruments."

The reviewers had some suggestions and comments, which appear to be addressed by the authors.